# Orthobiologic Management Options for Degenerative Disc Disease

**DOI:** 10.3390/bioengineering11060591

**Published:** 2024-06-10

**Authors:** Cezar Augusto Alves de Oliveira, Bernardo Scaldini Oliveira, Rafael Theodoro, Joshua Wang, Gabriel Silva Santos, Bruno Lima Rodrigues, Izair Jefthé Rodrigues, Daniel de Moraes Ferreira Jorge, Madhan Jeyaraman, Peter Albert Everts, Annu Navani, José Fábio Lana

**Affiliations:** 1Orthopedics, ABCOliveira Medical Clinic, São Paulo 03310-000, SP, Brazil; prolife.coluna@hotmail.com (C.A.A.d.O.); prolifecoluna@hotmail.com (B.S.O.); prolife.coluna@gmail.com (R.T.); 2Learning and Teaching Unit, Queensland University of Technology, Brisbane, QLD 4059, Australia; j3.reilly@qut.edu.au; 3Department of Orthopedics, Brazilian Institute of Regenerative Medicine (BIRM), Indaiatuba 13334-170, SP, Brazil; brunolr.ioc@gmail.com (B.L.R.); neurovirtual@gmail.com (I.J.R.); danfjorge@gmail.com (D.d.M.F.J.); josefabiolana@gmail.com (J.F.L.); 4Regenerative Medicine, Orthoregen International Course, Indaiatuba 13334-170, SP, Brazil; madhanjeyaraman@gmail.com (M.J.); everts@me.com (P.A.E.); annu@navani.net (A.N.); 5Department of Orthopaedics, ACS Medical College and Hospital, Dr MGR Educational and Research Institute, Chennai 600077, Tamil Nadu, India; 6Medical School, Max Planck University Center (UniMAX), Indaiatuba 13343-060, SP, Brazil; 7Medical Director, Le Reve, San Jose, CA 95124, USA; 8Chief Medical Officer, Boomerang Healthcare, Walnut Creek, CA 94598, USA; 9Medical School, Jaguariúna University Center (UniFAJ), Jaguariúna 13918-110, SP, Brazil; 10Clinical Research, Anna Vitória Lana Institute (IAVL), Indaiatuba 13334-170, SP, Brazil

**Keywords:** disc disease, orthobiologics, inflammation, orthopedics, regenerative medicine

## Abstract

Degenerative disc disease (DDD) is a pervasive condition that limits quality of life and burdens economies worldwide. Conventional pharmacological treatments primarily aimed at slowing the progression of degeneration have demonstrated limited long-term efficacy and often do not address the underlying causes of the disease. On the other hand, orthobiologics are regenerative agents derived from the patient’s own tissue and represent a promising emerging therapy for degenerative disc disease. This review comprehensively outlines the pathophysiology of DDD, highlighting the inadequacies of existing pharmacological therapies and detailing the potential of orthobiologic approaches. It explores advanced tools such as platelet-rich plasma and mesenchymal stem cells, providing a historical overview of their development within regenerative medicine, from foundational in vitro studies to preclinical animal models. Moreover, the manuscript delves into clinical trials that assess the effectiveness of these therapies in managing DDD. While the current clinical evidence is promising, it remains insufficient for routine clinical adoption due to limitations in study designs. The review emphasizes the need for further research to optimize these therapies for consistent and effective clinical outcomes, potentially revolutionizing the management of DDD and offering renewed hope for patients.

## 1. Introduction

Degenerative disc diseases (DDDs) encompass a wide and heterogeneous set of health conditions which can affect all musculoskeletal and nervous tissues along the spine [1]. DDDs are frequently associated with pain syndromes, radiculopathy, spondylosis, spondylolisthesis, stenosis, fractures, tumors, and osteoporosis [1]. DDD is linked to significant pain and disability, generating a major socioeconomic burden given its high global prevalence [2]. Patients often present pseudoradicular pain, mostly due to degenerative processes involving intervertebral discs (IVDs), facet joints, and soft tissues [3]. Pseudoradicular pain can also have less common causes such as infections, non-infectious inflammation, metabolic syndrome, and tumors [3]. The most obvious and common cause of radicular pain is likely compression of the nerve roots, which can arise from numerous intrinsic and extrinsic factors as well. This type of pain can also be accompanied by degenerative processes, infections, and secondary problems, including other diseases and injuries involving anatomical sites along the spine [4].

According to a systematic review [5], 266 million individuals (3.63% of the global population) suffer from degenerative spinal diseases (DSD) and lower back pain (LBP) every year, with prevalence rates highest in Europe (5.7%) and lowest in Africa (2.4%). Additionally, these conditions disproportionately affect low- and middle-income countries, which experience four times as many cases as high-income countries. Furthermore, global incidence rates reveal that 403 million people (5.5%) experience symptomatic disc degeneration, 103 million (1.41%) suffer from spinal stenosis, and 39 million (0.53%) are affected by spondylolisthesis each year. 

Many conventional management strategies have been studied and proposed for DDDs. For instance, physical therapy and non-steroidal anti-inflammatory drugs (NSAIDs) are common conservative treatments [6]. However, it must be emphasized that although NSAIDs and corticosteroids can effectively target pain, chronic use of these medications risks complications such as peptic ulcer disease, acute renal failure, and stroke/myocardial infarction [7,8]. In more severe injuries such as lumbar disc herniation, for example, these conventional management strategies may not suffice, ultimately forcing the patient to seek alleviation from orthopedic surgeons. Similarly, other alternatives, such as intradiscal electrothermal treatments, are not always effective. Arthrodesis is often recommended for discogenic back pain but there is still some controversy regarding this in the literature [6].

Given the shortcomings of traditional therapies, the demand for novel solutions for DDD has motivated the medical community to contribute to the expansion of medical biotechnology. Continuous research has promoted significant growth in the regenerative medicine market with the emergence of orthobiologics; derivates of endogenous molecules, cells, or tissues applied to injured tissue to prompt regeneration [9]. Some of the most popular contemporary orthobiologic alternatives commercially available are autologous solutions such as platelet-rich plasma (PRP), bone marrow-derived products and adipose tissue derivatives [9]. These materials trigger a wide set of biological responses that contribute to the amelioration of regeneration in areas where standard tissue healing is disrupted or difficult to achieve [10]. Although the biological mechanisms are complex, the aforementioned products share many similarities, especially in virtue of autocrine and paracrine signaling via release of several bioactive molecules [9,10,11,12].

DDDs are challenging and often present a multifactorial nature; therefore, physicians must be able to thoroughly evaluate the patient to correctly identify the root source of the problem. Physical examination and medical imaging methods are indispensable in order to avoid misdiagnosis. Delays between initial symptoms, correct diagnosis, and onset of treatment are also another important factor that must be avoided in order to achieve successful clinical outcomes [4].

The objective of this manuscript is to explain DDD pathophysiology and review the potential of the orthobiologics platelet-rich plasma, mesenchymal stem cells, and an orthobiologic adjunct material (hyaluronic acid) in the treatment of this spinal condition according to what has been documented in the literature. To achieve this, we searched the PubMed database and utilized forwards and backwards snowballing to identify both preclinical and clinical research on these orthobiologic interventions. Additionally, we searched two clinical trial repositories (ClinicalTrials.gov and EudraCT; accessed on 1 February 2024) for DDD to identify any current clinical trials assessing orthobiologic treatments. We synthesized our findings by firstly outlining the etipathogenesis of DDD, followed by preclinical evaluations of each orthobiologic and finally by presenting clinical trials on these orthobiologics.

## 2. Etiopathogenesis

The degenerative alterations and abnormalities that affect the spine involve bony structures and the IVD. In addition to age-related changes, these alterations may also be linked to traumatic, metabolic, toxic, vascular, infectious, and genetic factors [13,14]. However, there is also a significant incidence of chronic overload and sequelae of acute traumatic spine injuries [13]. Abnormal physical stress, even if not sufficient to cause fracture, can still harm bone and disc if maintained for a large amount of time [13]. At the cervical level, the distribution of axial load is normally responsible for degenerative alterations of C5-6 and C6-7 vertebrae in most cases; in the lumbosacral tract the most frequently affected sites are L4-5 and L5-S1 since these levels suffer the highest dynamic and static loads [15].

The IVD structurally consists of a fibrocartilaginous annulus fibrosus (AF) encasing a gel-like matrix termed the nucleus pulposus (NP). The NP matrix is maintained through proteoglycan and type II collagen synthesis by NP cells (NPCs). The interaction between these two extracellular matrix components imbues the NP matrix with hydroscopic properties, which is crucial for IVDs to be able to absorb compressive shocks during movement [16]. Any damage to the endplates due to these cyclic compressive loads marks the beginning of the cascade of degenerative disc disease as shown in Figure 1.

However, these same cells also produce matrix metalloproteinases (MMP) which degrade the NP ECM. The presence of anabolic signals, such as IGF-1, TGF, and bFGF, promotes ECM deposition [17], whereas various inflammatory cytokines (typically secreted following tissue injury) promote ECM degradation [18,19,20,21]. Therefore, if the cellular microenvironment shifts IVD cells to a catabolic state, degradation of the NP matrix will begin [22].

### 2.1. Age

Aging gradually increases the production of collagen, favoring collagen I production over collagen II production, which makes the IVD gradually more fibrous [23]. The anatomical demarcation between the nucleus pulposus and the annulus fibrosus becomes significantly diminished as these two regions merge [14]. The increase in collagen and collagen–proteoglycan binding reduces the availability of proteoglycan polar groups and therefore their capability to bind to water [24]. Consequently, desiccation renders the nucleus pulposus more solid and granular, increasing the risk of cracks and injuries not only to itself but to adjacent structures as well [23]. Senescence of IVD cells also has a fair share of culpability in these pathological developments, as it significantly reduces their ability to proliferate [25]. Additionally, senescent cells also contribute to degenerative progression by means of decreased anabolism and/or increased catabolic activity, reducing the tissue’s ability to compensate for net loss [25,26].

### 2.2. Genetics

Certain genetic factors are also partially responsible for DDDs in some cases, even more so in comparison to environmental factors. For instance, polymorphisms in the promoter region of the MMP-3 gene can accelerate degenerative alterations in the lumbar tract in elderly populations [27]. Similarly, patients with Trp2 and Trp3 variants in type IX collagen are also more susceptible to lumbar disorders because this mutation generates an unstable triple helix that is less resistant to mechanical stress [28]. Interestingly, variations in pro-inflammatory mediator genes are also at play in DDDs. A study shows that IL-1αT889 and IL-1βT3954, pathological alleles of the interleukin-1 gene, have been associated with disc bulging [29].

### 2.3. Nutrition

IVD cells obtain their nutrients from blood vessels in the peripheral soft tissue structures, relying on nutrient diffusion from capillaries across the cartilaginous endplate and disc matrix to the cells [30]. Insufficient or disrupted blood supply to these structures could be the main causative factor underpinning pathological progression [31]. Reduced delivery of nutrients leads to cell death and, ultimately, increases in oxidative stress markers [30,31]. Low oxygen levels and the acidic pH resulting from anaerobic metabolism impair the synthesis of proteoglycans and other proteins [30]. Furthermore, poor metabolic health also plays a significant role in cell nutrition, as is the case with that metabolic syndrome, which is known to disrupt standard cell activity [32].

### 2.4. Mechanobiology

Mechanical overload is frequently associated with spine conditions, especially when it comes to disorders affecting the lumbar discs [33]. The discs are structures designed to sustain and disperse mechanical forces. In fact, loading is a key component in human biology as physical stimuli trigger many developmental processes, including the regulation of matrix turnover [34]. Conversely, excessive loading harms these structures not only via physical damage but also by reducing gene expression of anabolic proteins, favoring a catabolic shift and inflammation [35].

Sudden and severe compressive forces can lead to fractures of the vertebral endplate, causing multiple problems. Although callus formation is a natural healing response, it can occlude blood vessels in the endplate and block the delivery of nutrients and oxygen for cells, disrupting the maintenance of the extracellular matrix [36]. Over time, damage to the endplate gradually leads to depressurization of the nucleus pulposus, applying more stress to the annulus fibrosus [37]. Consequently, this structure is no longer braced by the nucleus pulposus, generating greater interlaminar shear stress, delamination, and, ultimately, tearing of the annulus fibrosus [37]. The resulting bone marrow lesions are dense with newly formed nerve endings which are exposed to the inflammatory microenvironment and inadequately braced compressive forces (Figure 1). These newly-proliferated nerves are thought to be a major source of pain [38]. Destruction of cartilage from an endplate fracture, in turn, triggers an inflammatory response mediated by IL-1β, with the subsequent production of catabolic enzymes that attack matrix proteins [39]. Lastly, the matrix becomes vulnerable not only to unfavorable pH but to blood in the vertebral bodies as well, increasing the risk of an equivocal immunogenic response [40,41].

## 3. Orthobiologic Solutions

### 3.1. Platelet-Rich Plasma

Platelet-rich plasma is an orthobiologic derived from the patient’s own blood and has been researched in clinical settings for nearly 40 years [42]. Its preparation methods are diverse; however, all aim to concentrate the platelet count within the patient’s plasma sample before delivering it to the site of desired tissue regeneration [43]. Given the role of platelets in secreting growth factors and promoting regeneration of damaged tissue, PRP has many predicted functions at the degenerating disc. Additionally, PRP’s orthobiologic nature ensures that it will be non-immunogenic and carry no risk of disease transmission in comparison to xenobiologic or allobiologic treatments [44]. Given its long history and relatively non-invasive preparation protocol in comparison to bone marrow aspirates, an array of preclinical and clinical research has examined the effectiveness of PRP in treating DDD.

PRP application to cultured cells of the intervertebral disc, including NPCs and annular fibrosis cells (AFCs), has been investigated to provide preclinical evidence for its efficacy in treating DDD. Cultured NPCs from porcine and rabbit NPCs treated with PRP resulted in an increase in cell proliferation, proteoglycan synthesis [45], and increased transcription of extracellular matrix genes [46], respectively. In a model of immortalized human NPCs, LPS-induced inflammation was reversed by PRP administration [47]. These results mirror the anti-inflammatory effects seen from PRP administration to cultured human NPCs following IL-1β and TNF-α-induced inflammation.

The influence of PRP on NPC proliferation and differentiation is unclear. In one study, human NPCs derived from healthy donors and cultured with PRP display increased proliferation and the expression of chondrogenic genes [48]. This study also demonstrated a synergistic effect between PRP and 1 ng/mL of TGF-1β [48]. The findings of Mietsch et al. [49] instead conclude that TGF-1β alone induces chondrogenesis on human NPCs more strongly than PRP. Mietsch and colleagues [49] ultimately suggest that this discrepancy is due to other differences present within the cell media. While both studies demonstrated an induction of anabolism by PRP, other environmental variables likely determined whether this anabolic programming resulted in the mutually exclusive processes of cellular differentiation (i.e., chondrogenesis) or proliferation [50].

PRP administration to cultured AFCs has also been examined. Pirvu et al. [51] reported an increase in ECM synthesis and cellular proliferation following PRP administration to bovine AFCs. Complementary results were also generated in Hondke and colleagues’ [52] study of cultured human AFCs. Specifically, they found that PRP increased cell proliferation and ECM synthesis in AFCs. Lastly, when porcine AFCs also increase their cellular proliferation and ECM synthesis in response to PRP exposure [45]. Taken together, these studies demonstrate that PRP plasma also promotes anabolism in AFCs in vitro. In summary, PRP administration shifts cultured NPCs and AFCs to an anabolic, anti-inflammatory state ideal for regenerating the cellular and extracellular environment of the degenerating intervertebral disc.

Given these promising in vitro findings, it is not surprising that many studies have progressed to assessing the efficacy of PRP in reversing DDD using preclinical animal models. The first of such studies was performed by Nagae and colleagues [53] on nucleotomised rabbits. PRP treatment alone demonstrated no difference to control groups; however, when combined with a hydrogel scaffold, PRP treatment increased proteoglycan presence at the AF and halted further disc degeneration [53]. Using the same treatment combinations on a larger sample of nucleotomised rabbits revealed similar findings; only combined PRP impregnated into gelatin hydrogel microspheres increased proteoglycan gene transcript and decreased apoptosis [54]. These early experiments supported the notion that PRP administration needed to be appropriately scaffolded to allow for its facilitation of regeneration to occur in situ.

In contrast, a study examining the anatomical integrity of needle-punctured rat IVDs showed that PRP injection alone protected against further anatomical degradation, an effect that was amplified the earlier the PRP was supplied post-injury [55]. Additionally, Pirvu et al. [51] also examined PRP injection in three bovine IVDs with an AF defect. Following PRP injection, matrix synthesis increased at the defect site; however, this study lacked statistical power to detect significant differences between treatment groups.

These contrasting results again point to the synergistic nature of PRP treatment when combined with other agents. For example, Wang and colleagues’ analysis of treatments for needle-punctured rabbit IVDs showed only minor structural regeneration in PRP-treated groups; however, groups treated with combined PRP and bone marrow-derived mesenchymal stem cells had far superior disc regeneration when assessed histologically and by MRI [56]. In summary, PRP is an effective orthobiologic agent that reprograms degenerating IVD cells to an anabolic, anti-inflammatory state. Its effectiveness is also increased when paired with an appropriate scaffold or additional orthobiologic agent. A number of clinical trials (Table 1) have therefore examined lone PRP, or PRP combinations as a treatment for human DDD.

### 3.2. Mesenchymal Stem Cells

The previous interventions discussed rely on shifting pre-existing IVD cells into an anabolic state to facilitate disc regeneration. However, multipotent stem cells can also be delivered to the site of degeneration with the hope that they will differentiate and replenish the intervertebral disc population. The potential of MSCs as a DDD therapeutic was first demonstrated in in vitro experiments. MSCs, when co-cultured with NPCs extracted from degenerating discs, promote MSC differentiation into chondrogenic cells with NPC phenotypes, whilst also promoting ECM production [81,82]. MSCs also initiate anti-inflammatory, anti-apoptotic cues within the IVD microenvironment [83]

In addition to their regenerative potential (Figure 2), another therapeutic advantage of MSCs is the relative ease with which autologous MSCs can be sourced. Harvesting autologous IVD cells directly has limited therapeutic potential [84] and potentially further damages the vertebral column. In contrast, autologous or allogeneic MSCs can be sourced from a variety of non-vertebral tissues, including bone marrow [9], adipose tissue, and umbilical cord blood [85,86].

MSCs regenerate damaged IVDs through three mechanisms. Firstly, MSCs may directly differentiate into IVD cells, including AFCs [87] and NPCs [88], thus replenishing the cellular population. MSCs also alter the cellular microenvironment of existing IVD cells. Paracrine signaling following the release of growth factors and ECM components can shift IVD cells to an anabolic state [89]. Lastly, MSCs may also release immunomodulatory mediators to blunt destructive immune responses to the compromised IVD microenvironment [90].

While MSCs are highly effective in vitro, their ability to survive in situ is limited. The hypoxic, acidic environment of the degenerating disc [30] is often fatal for implanted MSCs [91,92]. Therefore, emerging research has examined strategies to differentiate MSCs prior to their implantation. For example, preconditioning MSCs with a hypoxic culture environment promotes their expression of NPC phenotypes [88,93]. Alternatively, hydrogels, including those composed of hyaluronic acid, can alter the cellular environment of MSCs, prompting their differentiation into NPCs [94,95]. There are a plethora of additional pre-conditioning strategies for MSCs which are currently being examined for their effectiveness in DDD; a comprehensive review of all of these strategies has been recently published by Ohnishi et al. [96] MSCs therefore represent a diverse variety of orthobiologic treatments for DDD.

### 3.3. Adjunct Materials—Hyaluronic Acid

Importantly, the efficacy of orthobiologics is heavily influenced by adjunct materials that accompany their injection at the degenerating disc. This section will illustrate the importance of adjunct materials by reviewing the use of hyaluronic acid (HA) in DDD. HA is an unsulfated glycosaminoglycan constituting a major component of the extracellular matrix [97]. Its hydroscopic properties, like other glycosaminoglycans, allow it to facilitate shock absorption within the intervertebral discs. Additionally, large HA fragments also function to inhibit neighboring extracellular receptors that induce inflammatory or nociceptive cascades [98]. Lastly, hydrogels have been widely developed from HA scaffolds, allowing for more customisable mechanical properties. Often, these HA hydrogels can be used as a vehicle to deliver other orthobiologic agents to the joint site [99]. HA is therefore a promising regenerative material for the treatment of degenerative disc disease.

In vitro studies have shown that HA has significant influence over cultured NPC function. Firstly, Alini et al. [100] demonstrated that a matrix of type I collagen and supplied HA stimulated the production of proteoglycans in both NPCs and AFCs. When HA is administered to cultured NPCs derived from human degenerating discs it stimulates mitophagy which has downstream protective effects against apoptosis and degradation of the ECM [101]. Similar ECM-protective effects of HA were demonstrated in a separate study examining the application of a HA hydrogel with fibroblast growth factor in both human and bovine NPCs [102]. Importantly, these regenerative effects are concentration-dependent in a non-linear manner; Gansau and Buckley [103] found that very high concentrations of HA would suppress rather than enhance collagen production in bovine chondrocytes. Isa et al. [104] stimulated cultured bovine NPCs with IL-1β to model the inflammatory conditions of disc degeneration. Cells that were supplied with a HA hydrogel again downregulated their expression of pro-inflammatory signals whilst also downregulating the production of neurotrophins, potentially providing a molecular basis for the analgesic function of HA.

HA has also been tested in many preclinical models of degenerative disc disease. Firstly, Isa et al. [105] expanded on their previous in vitro findings and demonstrated that HA alleviates pain in a rat model of disc degeneration. These behavioural findings were supported by a concurrent decrease in cFOS expression in the left dorsal horn of the spinal cord of HA-treated rats. Finally, a comparative proteomic analysis mirrored in vitro findings, demonstrating an anti-inflammatory shift in protein expression induced by HA exposure. The hydroscopic properties of HA hydrogels also demonstrated regeneration potential in a rabbit model of disc degermation; intra-articular injection with a HA hydrogel increased disc height and water absorption and overall decreased the severity of degermation as scored through MRI [106]. Similar findings were observed using a different HA hydrogel to treat a murine model of DDD [94]. Interestingly, a similar in vivo study on the goat degenerating disc demonstrated no advantage to injecting BMP-2/7 through a HA hydrogel when compared to BMP-2/7 injection alone [107]. In a study of explanted injured murine discs, HA hydrogels decreased the expression of IGFBP3, IFNa, and caspase 3, whilst upregulating the key ECM components HAPLN1 and aggrecan [108].

To summarise, hyaluronic acid could act as a key delivery vehicle for orthobiologics in the treatment of degenerative disc disease. Its FDA approval for the treatment of knee osteoarthritis [109] has led to extensive investigations of its efficacy in other joints, including the intervertebral discs. In vitro research on NPCs and explanted disc tissue consistently demonstrates HA’s potential to attenuate inflammation, regenerate the ECM and reduce pain reception. Preclinical animal models utilizing HA hydrogels are also effective in regenerating the degenerating disc; however, the specific contribution of HA to this regeneration is hard to characterize given that it is often used in combination with other growth factors or other orthobiologics. However, all studies demonstrated an excellent safety profile for intradiscal HA injection.

## 4. Clinical Evidence

Published scholarly literature and clinical trial repositories (ClinicalTrials.gov and EudraCT) were searched to identify completed clinical trials evaluating the use of the preceding orthobiologic tools in the treatment of DDD. The results of the 19 clinical trials analysed are briefly summarized in Table 1. For an overview of ongoing clinical trials on cell-based therapies for DDD, refer to the review by Binch et al. [110] and Table 2.

## 5. Discussion & Future Directions

Overall, current clinical data suggests that orthobiologic treatments, such as intradiscal PRP, HA, or MSC injection, have the potential to outperform conventional therapies. All studies demonstrated a sustained reduction in reported pain. Given that a majority of these studies only included patients who did not respond to conventional DDD therapies, these results are even more promising. However, as others have concluded [111], the current clinical data cannot be used to recommend orthobiologic treatment for DDD without further research. Additionally, very few studies make direct comparisons between different orthobiologic approaches. Given this limited evidence, it is not possible to directly compare or rank the efficacy of different orthobiologic approaches for different diverse patient populations at this time.

Firstly, most of these clinical studies are prospective in nature and lack control groups. These controls are important, as illustrated by one study, which found that most patients receiving PRP injection reported a decrease in pain at follow-up, but that this pain reduction was not significantly different than patients in the control group who received a sham injection. Additionally, the sample sizes of these trials are often very low, which limits the statistical power of this research. Lastly, very few studies [68] make direct comparisons between orthobiologic treatments, which means that clinicians cannot currently discern which orthobiologic treatment is best suited for their patient/s. Additionally, of the ongoing clinical trials identified in our review, none have utilized a combinatorial orthobiological approach. Future research should not be aimed at demonstrating the potential of orthobiologics in DDD, but rather aimed at demonstrating which combination of orthobiologics and treatment methods produce the most favourable outcomes.

The sheer number of potential therapeutic strategies combining cellular therapies with scaffolds makes this a daunting task [112]. This is further exacerbated by the long-standing inconsistencies present in preclinical animal models of DDD and orthobiologic treatments [110,113,114]. In order to navigate these issues, in vitro studies should first be performed to compare and assay confounding variables (e.g., stem-cell type, culturing conditions, presence or absence of growth factors). Then, culturing conditions which yield the most regenerative phenotype can be used to plan preclinical animal experiments. Furthermore, the intersection of orthobiologic therapies with other DDD management approaches, such as physical therapy and psychological support as part of holistic treatment strategies, must also be investigated to improve patient outcomes [115]. In doing so, clinical trials can be reserved for pre-optimized orthobiologic therapies.

## 6. Conclusions

In conclusion, while orthobiologic treatments such as intra-discal PRP, HA, or MSC injections show promise in surpassing conventional therapies, their efficacy in the management of DDD is yet to be conclusively determined. The current clinical data, characterized by a lack of controls and small sample sizes, limits the ability to definitively recommend these treatments. Future research should focus not only on validating the potential of orthobiologics but also on identifying the most effective combinations and methods. Addressing the inconsistencies noted in preclinical models and enhancing study designs will be crucial. Optimized, evidence-based orthobiologic approaches could then provide a breakthrough in DDD treatment, significantly reducing pain and improving life quality for patients.

## Figures and Tables

**Figure 1 bioengineering-11-00591-f001:**
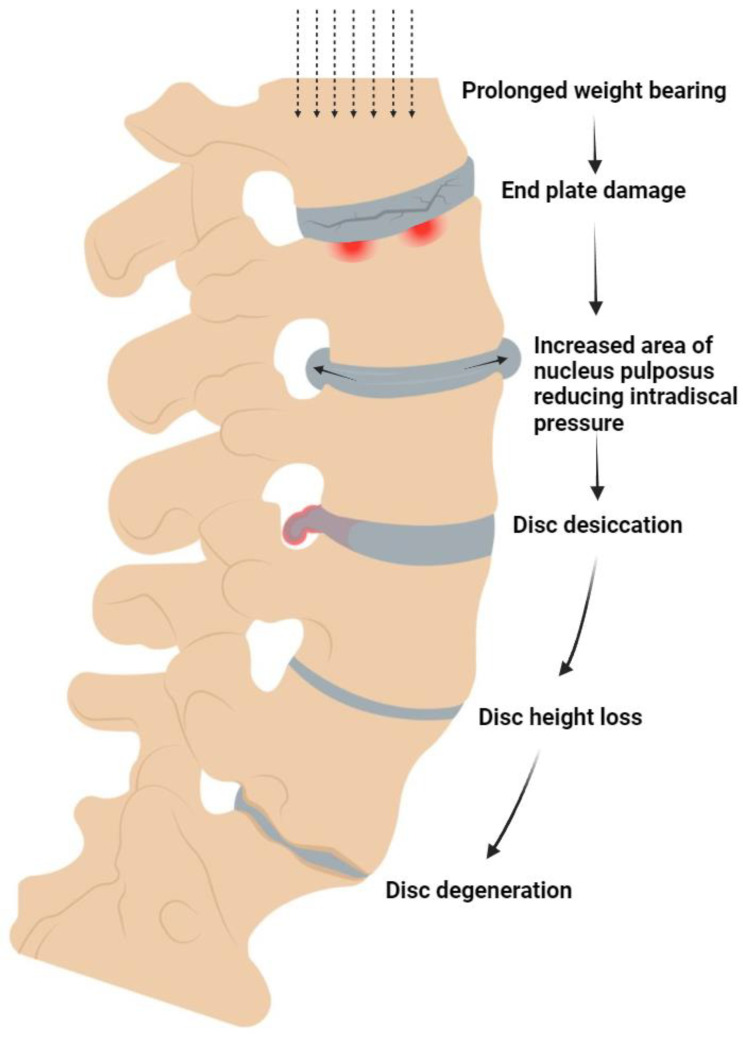
Pathophysiology of internal disc disruption secondary to vertebral end plate injury leading to disc degeneration.

**Figure 2 bioengineering-11-00591-f002:**
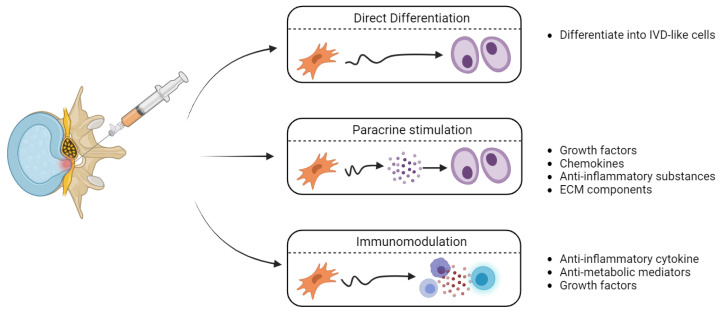
Mechanism of action of transplanted MSCs in the context of degenerative IVDD.

**Table 1 bioengineering-11-00591-t001:** Published Clinical Trials examining the effectiveness of intradiscal orthobiologic injections for degenerative disc disease. (BMMSC = bone marrow-derived MSCs, AMSC = adipose-derived MSCs, UCMSC = umbilical-cord-derived MSCs, PRP = platelet-rich plasma, HA = hyaluronic acid).

Treatment	Reference	Total Participant Number	Trial Design	Outcome
BMMSC	[57]	24	Randomized controlled trial—single intradiscal injection of BMMSCs or sham injection (unspecified anesthetic).	Reduced pain and degeneration at 12 months follow-up. Feasibility and safety confirmed.
BMMSC	[58,59,60]	26	Prospective, open-label nonrandomized trial—intradiscal injection of BMMSCs at one IVD (n = 13) or two adjacent IVDs (n = 13).	Reduced pain at 1,2, and 3 years follow up
BMMSC	[61]	10	Single treatment group—single intradiscal injection of BMMSCs.	Feasibility and safety confirmed. Reduced pain and disability at 3 months. IVD water content increased at 12 months.
BMMSC	[62]	10	Single treatment group—single intradiscal injections of BMMSCs followed by 2 weeks of hyperbaric oxygen therapy.	No pain reduction at 12 months follow up.
BMMSC	[63]	2	Single treatment group—single intradiscal injection of collagen scaffold soaked in BMMSCs.	Reduced pain and vacuum phenomenon (gas in IVD) at 24 months follow-up.
BMMSC	[64]	5	Single treatment group—single intradiscal injection of hypoxic-cultured BMMSCs.	No adverse outcomes. Improved mobility and strength reported for 4 patients at 4–6 year follow-up.
BMMSC	[65]	33	Single treatment group—single intradiscal injection of BMMSCs.	Safety confirmed. Pain reduction at 3–6 years follow-up. Of the 20 patients who underwent post-treatment MRI, 85% also had reduced disc bulge size.
BMMSC	[66,67]	11	Single treatment group—single lumbar intradiscal injection of BMMSCs embedded in tricalcium pohosphate.	Reduced pain and disability at 5 and 10 years follow-up. All imaged patients demonstrated lumbar fusion.
BMMSC & PRP	[68]	40	Multicenter randomized controlled trial—single intradiscal injection of BMMSCs, PRP, or saline (placebo control).	PRP reduced pain and improved function at 1 year follow-up when compared to placebo. BMAC reduced pain and improved function at 1 year follow-up when compared to placebo. No significant differences between PRP and BMMSC treatments were detected.
UCMSC	[69]	2	Single treatment group—single injection of UCMSCs.	No severe adverse events following treatment. Reduced pain at 24 months follow-up.
AMSC	[70]	15	Single treatment group—single injection of AMSCs.	No severe adverse events following treatment. Reduced pain and disability at 12 months follow-up
AMSC & HA	[71]	10	Single treatment group—single injection of AMSCs combined with a HA derivative.	No severe adverse events following treatment. Reduced pain at 1 year follow-up. Three patients demonstrated increased IVD water content in 1 year follow-up MRI.
PRP	[72]	47	Double-blind, randomized controlled trial. Single intradiscal injection of PRP (n = 29) or contrast agent (placebo control; n = 18).	Statistically significant pain reduction at 8 weeks follow-up for PRP treatment group when compared with placebo group.
PRP	[73]	22	Single treatment group—intradiscal injection of PRP in two IVDs (n = 10), three IVDs (n = 2) or five IVDs (n = 1).	Reduced pain and disability at 6 months follow-up.
PRP	[74]	26	Double-blind, randomized controlled trial. Single intradiscal injection of PRP (n = 18) or saline (placebo control; n = 8).	No significant differences in pain or disability reduction seen between PRP and placebo groups.
PRP	[75]	48	Double-blind, randomized controlled trial. Single intradiscal injection of PRP. Percutaneous intradiscal radiofrequency ablation.	Statistically significant reduction in pain and disability at 3 and 6 months follow-up; however, no statistically significant difference in pain/disability reduction between PRP and radiofrequency ablation groups.
PRP	[76,77]	16	Double-blind, randomized controlled trial. Single intradiscal injection of PRP releasate (n = 9) or betamethasone sodium phosphate (a corticosteroid; n = 7). Fifteen patients also received an additional, optional PRP injection 8 weeks after treatment.	Significant improvement in disability and walking ability in PRP releasate group when compared to corticosteroid group at 26 weeks follow-up. Both treatment groups had significant reduction in pain; however, no significant differences in pain reduction between groups were detected.
PRP	[78]	5	Single treatment group—single intradiscal injection of PRP.	Gradual pain and disability reduction up to and including at 1 year follow-up.
PRP	[79]	6	Single treatment group—single intradiscal injection of PRP.	Pain reduction at approximately monthly follow-ups for 6 months for all patients. Six months post MRI demonstrated structural improvements in disc anatomy for some patients.
PRP	[80]	14	Single treatment group—single intradiscal injection of PRP releasate.	No adverse effects observed following treatment. Statistically significant pain reduction at 1- and 6-month follow-ups. No significant differences detected in follow-up MRI T2 quantification.

**Table 2 bioengineering-11-00591-t002:** Ongoing Clinical Trials examining the effectiveness of orthobiologic treatments for degenerative disc disease not previously listed by Binch et al. (2021) [110]. (BMMSC = bone marrow-derived MSCs, AMSC = adipose-derived MSCs, UCMSC = umbilical-cord-derived MSCs, PRP = platelet-rich plasma).

Treatment	ClinicalTrials.gov ID	Total Participant Number	Protocol	Trial Status at March, 2024
PRP	NCT05287867	42 (28 treatment, 14 sham control)	Single-blind, randomized, placebo-controlled study. Two treatments, four weeks apart of intradiscal PRP (or sham injection).	Actively recruiting
PRP	NCT04816747	50 (estimated)	Single group assignment, single intradiscal PRP injection.	Not yet recruiting
PRP	NCT02983747	112 (estimated)	Randomised controlled trial (PRP intradiscal injection compared to thrice weekly oral NSAID (loxoprofen)).	Recruiting
UCMSC	NCT04414592	20 (estimated)	Single group assignment, single intradiscal UCMSCs injection.	Status unknown
AMSC	NCT05011474	4 (estimated)	Single group assignment (AMSC intradiscal injection enriched with the ECM protein matrilin-3).	Status unknown
BMMSC	NCT05066334	52 (estimated)	Randomized controlled trial (intradiscal injection of BMMSCs vs. sham control of local anesthesia).	Status unknown
BMMSC	NCT04759105	48	Randomized control trial (intradiscal injection of BMMSCs vs. sham control of local anesthesia).	Active, not recruiting
BMMSC	NCT04042844	99 (estimated)	Double-blind, randomized controlled trial (intradiscal injection of BMMSC vs. saline).	Actively recruiting
BMMSC	NCT04735185	106 (estimated)	Randomized controlled trial of single intradiscal injection (intradiscal injection of BMMSCs, methylprednisolone, or local anesthethic (bupivacaine) control).	Suspended (awaiting sponsor and FDA feedback)

## Data Availability

No new data generated.

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
