# Peer review of "Orthobiologic Management Options for Degenerative Disc Disease"

_bioengineering, 2024, doi:10.3390/bioengineering11060591_

Round 1

Reviewer 1 Report

Comments and Suggestions for Authors

Firstly, I want to congratulate you on the effort you have accomplished in conceptualizing, executing, and compiling this research. This discovery has significant implications for the community of healthcare professionals who often deal with this disorder and for their patients.

Nevertheless, the work possesses specific limits and formal flaws that must be corrected before it is published in the Journal.

In this manuscript, the authors performed a thorough examination of the current body of data on orthobiologic treatments for degenerative disc degeneration. The researchers examined papers that investigated the application of different regeneration methods, including stem cell therapy, growth factors, and tissue engineering, for the treatment of intervertebral disc degeneration.

I recommend that the article utilize a review format (Introduction, Materials and Methods, Results, Discussion, and Conclusions) or (Introduction to the topic, methodology used to select the studies to be reviewed, analysis of the results, and the conclusions and recommendations)

Introduction: You must provide a comprehensive and current overview of low back pain. It suggests naming the finite element method, which is widely used to analyze different pathologies related to degenerative disc diseases (https://doi.org/10.3390/

prosthesis5030065) or (https://doi.org/10.3390/jcm13092553

It does not clearly show the methodology used for the review. Explain how the articles were compiled and the parameters for the selection of analyzed cases.

 Table 1 is the most significant contribution of the research. Add a discussion section highlighting the importance of that table.

You must add a conclusion section.

Finally, I suggest adding some points to improve the paper

·         Include a comparative examination of various orthobiologic therapy approaches to have a better understanding of their relative efficacy and acceptability for diverse patient populations.

·         Please present a comprehensive analysis of current and concluded clinical trials concerning orthobiologic treatments for degenerative disc disease. This should include a thorough examination of their procedures, findings, and potential implications for future research. This point could be your discussion section.

·         Explore the integration of multidisciplinary approaches in orthobiologic management, such as combining regenerative techniques with physical therapy, pain management strategies, and psychological support for comprehensive patient care.

Comments on the Quality of English Language

No comments

Author Response

Dear reviewer, please see attachment.

Reviewer 2 Report

Comments and Suggestions for Authors

Gabriel Silva Santos et al. submitted an interesting review about DDD. The topic was less frequently discussed in the literature, but of a certain importance. Thus, the paper might arouse a certain impact in the field of bioengineering. Overall, this paper could be reconsidered for publication after a Major Revision. Please refer to the following comments:

1)     The Abstract contained ~160 words. Please consider to expand it to ~200 words to showcase more valuable information.

2)     Some critical statistics about DDD should be mentioned at the beginning of the Introduction.

3)     In Section 2, pathologies from the cellular and even molecular perspective could be introduced.

4)     At the beginning of Section 3, it was advisable to introduce non-orthobiologic solutions.

5)     It was inappropriate to show the reference number as “First author, year” in Table 1. Please consider to provide the exact information about “First author, year”, and identify the reference number as a new row.

6)     Line 262: Missing of subtitle number?

7)     Please define a separate Conclusion Section.

8)     The format of Reference needed double-check.

Author Response

Dear reviewer, please see attachment.

Round 2

Reviewer 2 Report

Comments and Suggestions for Authors

Thanks for your revision.